# Rapid bacterial identification and resistance detection using a low complexity molecular diagnostic platform in Zimbabwe

Tinashe Mwaturura[1,2*☉], Ioana D. Olaru[2,3☉], Gwendoline Chimhini[4], Mutsa Bwakura-Dangarembizi[4], Marcia Mangiza[5], Simbarashe Chimhuya[4], Belinda Sado[5], Jackie Katunga[5], Andrew Tarupiwa[6], Agnes Juru[6], Tapfumaneyi Mashe[6], Christopher Pasi[5], Veronicah Chuchu[7], Seyi Gansallo[7], Birgitta Gleeson[7], Felicity Fitzgerald[1,8‡], Cecilia Ferreyra[7‡], Katharina Kranzer[1,2,9‡]

1 The Health Research Unit, Biomedical Research and Training Institute, Harare, Zimbabwe, 2 Clinical Research Department, London School of Hygiene and Tropical Medicine, London, United Kingdom, 3 Institute of Medical Microbiology, University Hospital Münster, Münster, Germany, 4 Department of Child, Adolescent and Women's Health, University of Zimbabwe Faculty of Medicine and Health Sciences, Harare, Zimbabwe, 5 Sally Mugabe Central Hospital, Harare, Zimbabwe, 6 National Microbiology Reference Laboratory, AMR unit, Harare, Zimbabwe, 7 Foundation of Innovative Diagnostics, Medical Affairs Department, Geneva, Switzerland, 8 Department of Infectious Diseases, Faculty of Medicine, Imperial College London, London, United Kingdom, 9 Division of Infectious Diseases and Tropical Medicine, LMU University Hospital, LMU, Munich, Germany

☉ These authors contributed equally to this work.
‡ FF, CF and KK also contributed equally to this work.
* tinashemwaturura@gmail.com

## Abstract

### Background

Sepsis is a major cause of mortality in low-resource settings. Effective microbiological culture services are a bottleneck in diagnosis and surveillance.

### Aim

We aimed to evaluate the performance of the BIOFIRE FILMARRAY Blood Culture Identification 2 (BCID2, bioMérieux) assay in a low-resource setting laboratory in comparison to standard practice.

### Methods

This five month prospective validation study included all positive blood cultures collected at Sally Mugabe Central Hospital, Harare, Zimbabwe. BCID2 testing was done in parallel to standard phenotypic procedures and resistance testing. Reference identification was performed using mass spectrometry or whole genome sequencing. Only samples with available reference standard results were included in the analysis. Data captured on paper-based forms was entered into electronic case report forms (ODK Collect). Specificity and sensitivity for BCID2 were calculated in comparison to the reference standards, with performance measures calculated using the Wilson score. Biomedical scientists using BCID2 completed a system usability survey (SUS).

**Data availability statement:** Data and the codebook for the study have been uploaded as Supporting Information.

**Funding:** This work was funded through three grants awarded to the Foundation of Innovative Diagnostics. These grants include the Federal German Government's KfW Development Bank (Grant number: KFW-TBBU02); the Ministry of Foreign Affairs of the Government of the Netherlands (Grant number: NL-GRNT05) and the UK Department of Health and Social Care as part of the Global AMR Innovation Fund (GAMRIF) (Grant number: UKH-GAMR0). GAMRIF is a One Health UK aid fund that supports research and development around the world to reduce the threat of antimicrobial resistance in humans, animals and the environment for the benefit of people in low- and middle-income countries (LMICs). The funders had no role in study design, data collection and analysis, decision to publish, or preparation of the manuscript. The views expressed in this publication are those of the authors and not necessarily those of the funders.

## Results

Positive results were recorded in 780/2,023 (38.5%) blood cultures, within which 377 (48.3%) had reference results and so were included in analysis. Neonatal samples were most frequent (182, 48.3%), then paediatric (150, 39.8%), then adults (18, 4.8%) and unknown (27, 7.2%). Specificity exceeded 95% throughout. Sensitivity ranged from 50% (*A. calcoaceticus-baumanii* complex, *Proteus* spp.) to 100% (*S. pneumoniae*, *Salmonella* spp). Using BCID2, CTX-M was detected in 111/175 (74.5%) Enterobacterales, from which 5/111 also had NDM and VIM detected. NDM-5 was detected in 2/5 NDM samples using sequencing. In total 3/23 *S. aureus* isolates were methicillin resistant, from which one was confirmed using phenotypic antimicrobial susceptibility testing. Usability was good (SUS score = 79.5).

## Conclusion

Rapid molecular tests have potential to improve turn-around time and quality of sepsis diagnostics. However, specific work-flows are critical to supplement molecular tests with minimal phenotypic tests for optimal clinical decision-making.

## Introduction

Although low- and middle-income countries (LMIC) bear a disproportionately high burden of infectious diseases [1], they often face significant constraints in accessing high-quality diagnostics [2] with more than half of the world's population estimated to have limited or no access to diagnostic services [3]. Additionally, in LMIC settings where diagnostics are available, turnaround times are often too long to impact on patient care [4,5].

Bloodstream infections (BSI) and sepsis are leading causes of morbidity and mortality posing significant public health challenges particularly in low-resources settings [1]. An estimated 48.9 million sepsis cases occurred in 2017 resulting in 11 million deaths and accounting for 19.7% of global deaths [1]. Timely, effective antimicrobial treatment is crucial, as delays in treatment initiation are associated with poorer outcomes, particularly in vulnerable patients such as neonates and children [6]. Conventional culture-based methods for blood cultures typically provide results 48 hours post-detection of positive growth and require considerable infrastructure, staff expertise and quality management. Consequently, in LMIC, high quality culture-based microbiological services are largely confined to tertiary and private hospitals and research settings. Expanding access to laboratory testing by simplifying procedures and reducing delays in species identification and antimicrobial susceptibility testing are critical to effectively manage patients with sepsis and improve patient outcomes.

Conventional microbiology traditionally relied on using phenotypic, manual methods for pathogen identification and antimicrobial susceptibility testing. With technological advances, high income countries have moved to using more rapid methods such as mass-spectrometry or molecular-based methods for pathogen identification and introduced automatisation in antimicrobial susceptibility testing. These changes resulted in reduced turnaround times and improved reliability and overall quality of results. Costs for these instruments and their consumables remain high and their operation requires advanced technical skills thus making them often unaffordable for low-resource settings. Multiplex PCR assays for the identification of bacterial and fungal pathogens and detection of resistance genes from positive blood cultures show promise in reducing the time to results and simplifying laboratory workflows. Rapid detection of resistance is critical given the high burden of AMR and its associated

mortality particularly among the most vulnerable in LMIC [7–9] and is considered a research priority for AMR by the WHO [10].

A recent systematic review including nine studies from the US, Europe, Australia and Asia showed high sensitivity and specificity of the BIOFIRE FILMARRAY Blood culture Identification 2 (BCID2, bioMérieux, Marcy l'Etoile, France) panel for species identification compared to culture-based methods [11]. Sensitivity was overall lower for detection of resistance [11]. Only one of the nine studies was conducted in a low-income setting (India) and included a very limited number of positive blood cultures (n=30) [12].

The aim of this study was to evaluate the real-life microbiological performance of the BCID2 assay for identifying bacteria and detecting resistance genes in a large governmental referral hospital in Zimbabwe, through comparison with results of standard phenotypic and sequencing methods.

## Methods

### Study setting and participant inclusion

This prospective single-centre validation study was conducted at Sally Mugabe Central Hospital (SMCH), Harare, Zimbabwe, a governmental referral centre with 1650 beds. The hospital serves more than half of the population of Harare and hosts the largest neonatal intensive care unit in the country. All patients from the neonatal unit, paediatrics and adult wards with suspected blood stream infections who had positive blood cultures according to routine clinical and laboratory procedures between the 7th of June to the 10th of October 2022 were included in the study.

### Blood culture diagnostics

Clinicians collected blood cultures in the wards according to the hospital's standard operating procedures. Clinicians underwent regular blood culture collection technique training to reduce contamination and collect blood volumes required for the neonatal unit, paediatrics and adult blood culture bottles (1 – 2 ml, 3ml and 8 – 10ml, respectively). Blood culture samples were delivered to the laboratory, which was on site, in less than 4 hours. Blood culture bottles (Bactec Plus aerobic; BD) were incubated in a BD Bactec Fx instrument at 37°C. Bottles that flagged positive were removed and one drop (40µl) was streaked onto MacConkey, chocolate and blood agar plates. Incubation was carried out at 37°C for a maximum of 48h. Gram stains were performed from positive sub-cultures. If in stock, conventional phenotypic tests were used for pathogen identification, including bacitracin and optochin disc, catalase, coagulase, bile aesculin, indole, citrate, oxidase and analytical profile index (API) 20E dependent on availability. All positive sub-cultures were inoculated in tryptone soya broth prepared with 20% glycerol (TSB) and stored at -20 degrees for reference testing. A reference test, also referred to as a gold standard test, is used to discriminate patients with blood stream infections and those without evidence of blood stream infections.

### Multiplex PCR testing

The multiplex PCR testing was conducted using the BCID2 which is a fully automated microbiological diagnostic assay that can detect 11 Gram-positive bacteria, 15 Gram-negative bacteria, 7 fungal pathogens, and 10 antimicrobial resistance genes [13]. Testing was performed according to the manufacturer's instructions. Briefly, a hydration solution was loaded into the multiplex PCR panel pouch, and 0.2ml of the positive blood culture broth were mixed with the provided sample buffer. This mixture was added to the pouch and loaded onto the instrument which conducted nucleic acid extraction and multiplex nested PCR. A result was

obtained within 60 min. For samples with invalid results or errors, repeat testing on the multi-plex PCR panel was performed. Results were entered into a custom-made electronic database (Open-Data-Kit, ODK).

## Phenotypic identification and antimicrobial susceptibility testing

The reference testing was conducted from 30/11/2022 to 30/05/2023 using the stored positive sub-culture isolates inoculated in TSB. Blood cultures for which both the BCID2 panel and the initial phenotypic work-up identified coagulase negative *Staphylococci* were not further investigated. For reference testing, all other positive sub-cultures were defrosted and inoculated on MacConkey, chocolate, and blood agar plates. Incubation was carried out at 37°C and ambient conditions for a maximum of 48h. Bacterial isolates were identified using the VITEK MS (MALDI-TOF) mass spectrometry microbial identification system (bioMérieux) at the National Reference Laboratory in Zimbabwe. Suspected pathogens were shipped to IHMA Europe Sàrl, Switzerland, Ares Genetics GmbH, Austria and Microbes Ng, United Kingdom and identified using whole genome sequencing with Illumina NovaSeq Platforms and MALDI-TOF Microflex LT. Antimicrobial susceptibility testing was performed using broth microdilution and interpreted according to Clinical Laboratory Standards Institute (CLSI) version 29 [14].

## Data management and statistical analysis

**Data collection.** Data were collected on paper-based forms and entered into electronic case report forms using the ODK software. Range restrictions and dropdown menus were used to minimise data entry errors. No patient details were extracted from the laboratory records except for the department where the blood culture was collected: neonatal unit, paediatrics and adults.

MALDI-TOF or whole genome sequencing (if a MALDI-TOF result was not available) were considered as reference for identification. Phenotypic antimicrobial susceptibility testing conducted by the reference laboratories were considered as the reference standard for resistance determination.

**Preparation of data for analysis.** Only samples with available reference standard results were included in the analysis. A BCID2 result was considered a true positive (TP) or true negative (TN) when it agreed with the result from the reference method. A result was considered a false positive (FP) or false negative (FN) when it disagreed with the results from the comparator methods.

**Data analysis.** Sensitivity was calculated as 100 x TP/(TP+FN) while specificity was calculated as 100 x TN/(TN+FP). Sensitivities and specificities were calculated based on the pathogen species. If another pathogen was detected/ isolated from the sample, then the sample was considered negative for the purpose of the calculation. If a stored culture failed to grow on re-culture, the BCID2 result of the sample was not taken into consideration for the sensitivity and specificity calculation. The exact binomial two-sided 95% confidence intervals (95%CI) were calculated for performance measures according to the Wilson score method. When the BCID2 identified a positive blood culture as *Staphylococci* or *S. epidermidis* and the phenotypic work-up in the routine laboratory indicated coagulase negative staphylococci, the isolates were not further investigated and hence sensitivity and specificity for coagulase negative staphylococci was not determined.

## BIOFIRE BCID2 panel system usability

A total of five trained laboratory scientists who operated the Biofire instrument completed a usability survey. The survey was conducted from 11/10/2022 to 14/10/2022. The 10-item

System Usability Scale (SUS) is an assessment of product usability applicable to many types of technologies [15,16]. The survey questions were programmed onto ODK and were administered to the laboratory scientist by a research assistant. The SUS is a Likert scale on which respondents indicate the degree of agreement and disagreement with a statement on a 5 point scale with 1=strongly disagree and 5=strongly agree. The SUS statements cover a variety of aspects of system usability, including the need for support and training, system complexity, consistency and ease of use. SUS scores are calculated to range from 0 to 4 with 0=strongly disagree and 4=strongly agree. Statements 1,3,5,7 and 9 (positively worded statements) were derived from the average score minus 1 whilst SUS sores for statements 2,4,6,8 and 10 (negatively worded statements) were derived from 5 minus the average score. The total scores were multiplied by 2.5 to convert the range of possible values from 0 to 40 to 0 to 100 [15,16]. Existing literature suggest that a system with a SUS score above 68 is considered above average usability [15,16]. SUS yields a single number representing a composite measure of the overall usability of the system under study.

### Ethical considerations

Ethical approval for this work was obtained from the Medical Research Council of Zimbabwe [MRCZ/A/2850] and the Institutional Review Boards of the Biomedical Research and Training Institute [AP170/2021]. Individual consent from patients was not sought as any patient admitted at SMCH with clinical indications for a blood culture to be taken had a blood culture taken as per standard clinical care. The molecular work-up of the blood culture was done in parallel with the phenotypic standard of care work-up. Clinicians were not informed about the molecular results and no clinical decision was based on the molecular results.

Written informed consent was obtained from laboratory scientists who participated in the usability survey.

## Results

Among the 2023 blood cultures collected, 780 (38.5%) flagged positive (S1 Fig). Of those 86 (11.0%) had no growth on culture. Using BCID2, 190 (24.4%) positive blood cultures had no organism detected, 467 (59.9%) had one, 86 (11.0%) had two and 37 (4.7%) had three or more organisms detected (S2 Fig). The maximum number of organisms detected by BCID2 was six. In total 772 organisms were detected from the 590 BCID2 positive samples. Coagulase negative staphylococci were present as the only organism in 319/780 (40.9%) blood cultures identified by BCID2 and the phenotypic identification methods used at SMCH laboratory. These organisms underwent no further work-up and were not included in analysis.

Of the 190 samples where no organisms were detected by BCID2, 68 were also negative by culture. Reference test results were available for 377 samples which were further analysed. Of these, 182 (48.3%) were from the neonatal unit, 150 (39.8%) from paediatrics, 18 (4.8%) from adults and for 27 (7.2%) the department was unknown (Fig 1).

### Identification of organisms using reference methods

Using reference methods, 168/377 (44.6%) blood cultures were positive for Enterobacterales. The most frequently isolated Enterobacterales were *Klebsiella pneumoniae* in 116 (30.8%) of samples, *Escherichia coli* in 23 (6.1%), *Enterobacter cloacae* complex in 13 (3.4%), *Klebsiella oxytoca* in 12 (3.2%), and *Salmonella* spp in 8 (2.1%). Among the 182 blood cultures from neonates, 85 (46.7%) grew *K. pneumoniae* in culture. Other Gram-negatives isolated were *Acinetobacter calcoaceticus-baumanii* complex from 8 (2.1%) blood culture samples, *Pseudomonas aeruginosa* from 2 (0.5%), and *Neisseria meningitidis* in 1 (0.3%). Among

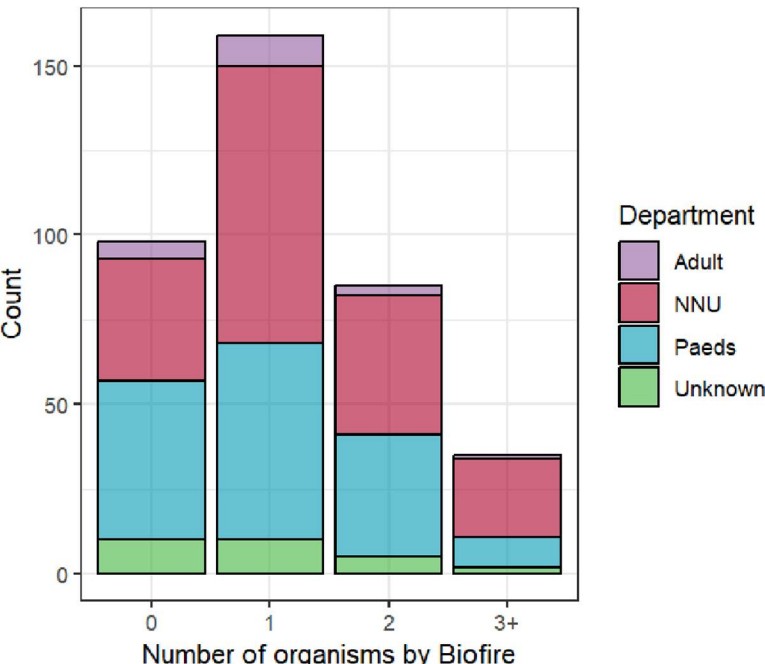

**Fig 1. Number of organisms identified by BCID2 in blood cultures showing growth on subculture.**

Gram-positives, *Staphylococcus aureus* was isolated from 14 (3.7%) samples, *Streptococcus agalactiae* from 11 (2.9%), *Enterococcus faecalis* from 29 (7.7%), *E. faecium* from 12 (3.2%), and *S. pneumoniae* from 1 (0.3%) sample. Organisms identified using the BCID2 panel are shown in Table 1.

## Performance of the BCID2 panel for bacterial identification

The BCID2 panel detected organisms in 279/377 (74%) blood culture samples either as single pathogens in 129/279 (46.2%) samples or alongside other organisms in 107/279 (38.4%). From 43/279 (15.4%) samples, only likely contaminants were detected (Fig 2). The diagnostic accuracy of the BCID2 panel is shown in Tables 2 and 3. Overall specificity was high exceeding 95% for all organisms. Sensitivity was variable ranging from 50% for *A. calcoaceticus-baumanii* complex and *Proteus* spp. to 100% for *S. pneumoniae*, *Salmonella* spp, *S. marcescens*, *N. meninigitidis* and *P. aeruginosa* although for the latter the number of isolates was generally very low. For, *K. pneumoniae* group, which was the organism most frequently identified, sensitivity was 84% and specificity 98%. Organisms identified in false positive and false negative samples are shown in S3 Table and S4 Table.

## Antimicrobial susceptibility testing results

Antimicrobial susceptibility testing results were available for 118/128 (92.2%) *Klebsiella* spp. isolates. The prevalence of resistance to ceftriaxone, ciprofloxacin and gentamicin was 90%, 37% and 40%, respectively (Fig 3). Only two isolates (1.7%) were resistant to meropenem, and no amikacin resistance was recorded. For *E. coli*, the prevalence of third-generation cephalosporin resistance was 2/23 (8.7%) while 7/25 (28%) were resistant to ciprofloxacin and 4/25 (16%) to gentamicin. For Gram-positives, only one *S. aureus* isolate was methicillin resistant and none of *the E. faecium* had vancomycin resistance.

**Table 1. Organisms identified using the BCID2 panel according to department[*].**

| BCID2 organism identification | Overall N = 377 | Neonatal unit N = 182 | Paediatrics N = 150 | Adult N = 18 | Unknown N = 27 |
|---|---|---|---|---|---|
| *Escherichia coli* | 25 (6.6%) | 12 (6.6%) | 9 (6.0%) | 4 (22%) | 0 (0%) |
| *Klebsiella pneumoniae* group | 103 (27%) | 78 (43%) | 18 (12%) | 0 (0%) | 7 (26%) |
| *Klebsiella oxytoca* | 16 (4.2%) | 8 (4.4%) | 8 (5.3%) | 0 (0%) | 0 (0%) |
| *Enterobacter cloacae* complex | 16 (4.2%) | 7 (3.8%) | 7 (4.7%) | 2 (11%) | 0 (0%) |
| *Serratia marcescens* | 4 (1.1%) | 3 (1.6%) | 1 (0.7%) | 0 (0%) | 0 (0%) |
| *Proteus* spp. | 2 (0.5%) | 1 (0.5%) | 1 (0.7%) | 0 (0%) | 0 (0%) |
| *Salmonella* spp. | 9 (2.4%) | 2 (1.1%) | 6 (4.0%) | 0 (0%) | 1 (3.7) |
| *Acinetobacter calcoaceticus-baumannii* complex | 7 (1.9%) | 3 (1.6%) | 2 (1.3%) | 1 (5.6%) | 1 (3.7%) |
| *Pseudomonas aeruginosa* | 3 (0.8%) | 1 (0.5%) | 2 (1.3%) | 0 (0%) | 0 (0%) |
| *Stenotrophomonas maltophilia* | 3 (0.8%) | 0 (0%) | 2 (1.3%) | 1 (5.6%) | 0 (0%) |
| *Neisseria meningitidis* | 2 (0.5%) | 0 (0%) | 2 (1.3%) | 0 (0%) | 0 (0%) |
| *Haemophilus influenzae* | 1 (0.3%) | 0 (0%) | 1 (0.7%) | 0 (0%) | 0 (0%) |
| Coagulase-negative staphylocci | 124 (33%) | 60 (33%) | 51 (34%) | 6 (33%) | 7 (26%) |
| *Staphylococcus aureus* | 23 (6.1%) | 10 (5.5%) | 11 (7.3%) | 0 (0%) | 2 (7.4%) |
| *Streptococcus* spp | 15 (4.0%) | 5 (2.7%) | 9 (6.0%) | 0 (0%) | 1 (3.7%) |
| *Streptococcus agalactiae* | 20 (5.3%) | 16 (8.8%) | 3 (2.0%) | 0 (0%) | 1 (3.7%) |
| *Streptococcus pneumoniae* | 3 (0.8%) | 0 (0%) | 3 (2.0%) | 0 (0%) | 0 (0%) |
| *Enterococcus faecalis* | 42 (11%) | 27 (15%) | 9 (6%) | 3 (17%) | 3 (11%) |
| *Enterococcus faecium* | 25 (6.6%) | 15 (8.2%) | 10 (6.7%) | 0 (0%) | 0 (0%) |
| *Listeria monocytogenes* | 1 (0.3%) | 1 (0.5%) | 0 (0%) | 0 (0%) | 0 (0%) |
| *Candida albicans* | 9 (2.4%) | 2 (1.1%) | 5 (3.3%) | 0 (0%) | 2 (7.4%) |
| *Candida parapsilosis* | 2 (0.3%) | 0 (0%) | 1 (0.7%) | 0 (0%) | 1 (3.7%) |
| *Candida (Nakaseomyces)glabrata* | 1 (0.3%) | 0 (0%) | 0 (0%) | 1 (5.6%) | 0 (0%) |
| Other *Candida* spp | 3 (0.8%) | 0 (0%) | 1 (0.7%) | 1 (5.6%) | 1 (3./) |

[*]Only organisms with a definitive reference identification are included

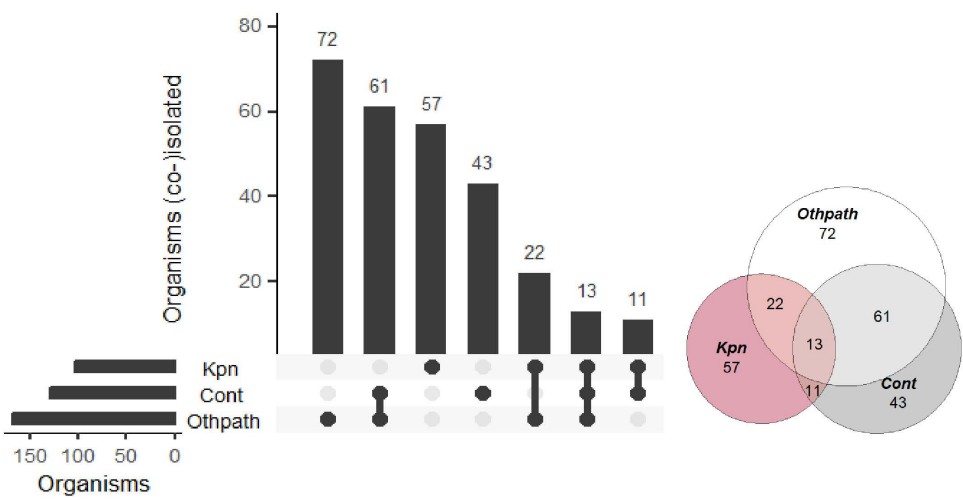

**Fig 2. Pattern of mono and polymicrobial blood culture samples as per BCID2 assay.** *Cont – likely contaminants, Kpn – K. pneumoniae, Otherpath – other likely pathogen.*

**Table 2. Performance summary of the BCID2 panel versus the comparator for all samples with a reference result available.**

| Organism | Isolates detected: BCID2/comparator | No. of results BCID2/ comparator | | | | Sensitivity TP/ (TP+FN) (%) | 95%CI | Specificity: TN/ (TN+FP) (%) | 95%CI |
|---|---|---|---|---|---|---|---|---|---|
| N=377 | | TP | FP | FN | TN | | | | |
| **Gram-positive** | | | | | | | | | |
| *E. faecalis* | 42/29 | 26 | 16 | 3 | 332 | 90 | 74–96 | 95 | 93–97 |
| *E. faecium* | 25/12 | 10 | 15 | 2 | 350 | 83 | 55–95 | 96 | 93–97 |
| *S. aureus* | 23/14 | 13 | 10 | 1 | 353 | 93 | 69–99 | 97 | 95–98 |
| *S. agalactiae* | 20/11 | 10 | 10 | 1 | 356 | 91 | 62–98 | 97 | 95–99 |
| *S. pneumoniae* | 3/1 | 1 | 2 | 0 | 374 | 100 | 21–100 | 99 | 98–100 |
| **Gram-negative** | | | | | | | | | |
| *A. calcoaceticus-baumannii* complex | 7/8 | 4 | 3 | 4 | 366 | 50 | 22–78 | 99 | 98–100 |
| *E. cloacae* complex | 16/13 | 11 | 5 | 2 | 359 | 85 | 58–96 | 99 | 97–99 |
| *E. coli* | 25/23 | 20 | 5 | 3 | 349 | 87 | 68–95 | 99 | 97–99 |
| *Salmonella* spp. | 9/8 | 8 | 1 | 0 | 368 | 100 | 68–100 | 100 | 98–100 |
| *K. oxytoca* | 16/12 | 11 | 5 | 1 | 360 | 92 | 65–99 | 99 | 97–99 |
| *K. pneumoniae* complex | 103/116 | 98 | 5 | 18 | 256 | 84 | 77–90 | 98 | 96–99 |
| *Proteus* spp. | 2/4 | 2 | 0 | 2 | 373 | 50 | 15–85 | 100 | 99–100 |
| *S. marcescens* | 4/3 | 3 | 1 | 0 | 373 | 100 | 44–100 | 100 | 99–100 |
| *N. meningitidis* | 2/1 | 1 | 1 | 0 | 375 | 100 | 21–100 | 100 | 99–100 |
| *P. aeruginosa* | 3/2 | 2 | 1 | 0 | 374 | 100 | 34–100 | 100 | 99–100 |
| *S. maltophilia* | 3/3 | 2 | 1 | 1 | 373 | 67 | 21–94 | 100 | 99–100 |

One of each L. monocytogenes and H. influenzae were detected by BCID2 but were not grown in culture. TP: true positives (number of organisms identified correctly by the BCID2 and which were detected with the reference method); FP: false positives (number of organisms which were detected by the BCID2 but not detected according to the reference method); FN: false negatives (number of organisms which were not detected by BCID2 but were detected by the reference method); TN: true negatives (number of organisms which were not detected by BCID2 and which were not present according to the reference method). TP, FP, FN, TN, sensitivities, and specificities were calculated based on the specific target organism. Samples in which a different species was detected/ isolated were considered negative for the purposes of the calculation.

## Detection of resistance genes by BCID2

A total of 175 Enterobacterales were detected using the BCID2 panel from 149 blood culture samples (Table 1). Of those, 111/149 (74.5%) had a concomitant detection of class A extended-spectrum β-lactamases (CTX-M). This included four samples where in addition to CTX-M, NDM was detected and one sample with New Delhi metallo-beta-lactamase (NDM) and Verona Intergron-encoded Metallo-beta-lactamase (VIM). Among the five blood culture samples with an NDM detection by BCID2, two had sequencing results and in both NDM-5 was detected. Among the 103 samples where *K. pneumoniae* group was detected, in 93/103 (90.3%) CTX-M was present. The prevalence of CTX-M using the BCID2 panel was 15/16 (93.8%) for *K. oxytoca* and 13/25 (52%) for *E. coli*. No samples had Imipenemase Metallo-beta-lactamase (IMP), *Klebsiella pneumoniae* Carbapenemase (KPC) or Oxacillinase-48-like beta lactamase (OXA-48-like) genes detected. An additional three samples had a CTX-M without Enterobacterales being detected by the BCID2 panel. One sample in which a *K. oxytoca* was detected had a mobilised Colistin resistance (mcr-1) gene by BCID2. From two of these samples, *E. coli* and *K. oxytoca* were isolated in culture. Vancomycin resistant genes (*Van*A or *van*B) were not detected in any of the samples. *S. aureus* was detected in 23 samples, of which three had Methicillin resistance genes (*mec*A/C and MREJ) and were thus Methicillin-resistant *Staphylococcus aureus* (MRSA). Of those, two underwent phenotypic antimicrobial susceptibility testing and one was confirmed as MRSA.

**Table 3. Performance summary of the BCID2 panel versus the comparator for monomicrobial samples with a reference result available.**

| Organism | Isolates detected: BCID2/comparator | No. of results BCID2/comparator | | | | Sensitivity TP/(TP+FN) (%) | 95%CI | Specificity: TN/(TN+FP) (%) | 95%CI |
|---|---|---|---|---|---|---|---|---|---|
| N=250 | | TP | FP | FN | TN | | | | |
| **Gram-positive** | | | | | | | | | |
| E. faecalis | 21/13 | 10 | 11 | 3 | 226 | 77 | 50–92 | 95 | 92–97 |
| E. faecium | 8/3 | 1 | 7 | 2 | 240 | 33 | 6–79 | 97 | 94–99 |
| S. aureus | 11/7 | 7 | 4 | 0 | 243 | 100 | 65–100 | 98 | 96–99 |
| S. agalactiae | 5/1 | 0 | 5 | 1 | 244 | 0 | 0–79 | 98 | 95–99 |
| **Gram-negative** | | | | | | | | | |
| A. calcoaceticus-baumannii complex | 1/3 | 0 | 1 | 3 | 246 | 0 | 0–56 | 100 | 98–100 |
| E. cloacae complex | 5/2 | 1 | 4 | 1 | 244 | 50 | 9–91 | 98 | 96–99 |
| E. coli | 6/5 | 4 | 2 | 1 | 243 | 80 | 38–96 | 99 | 97–100 |
| K. oxytoca | 13/9 | 9 | 4 | 0 | 237 | 100 | 70–100 | 98 | 96–99 |
| K. pneumoniae complex | 81/86 | 78 | 3 | 8 | 161 | 91 | 83–95 | 98 | 95–99 |
| Salmonella spp. | 1/1 | 1 | 0 | 0 | 249 | 100 | 21–100 | 100 | 98–100 |
| S. marcescens | 2/2 | 2 | 0 | 0 | 248 | 100 | 34–100 | 100 | 98–100 |
| N. meningitidis | 1/1 | 1 | 0 | 0 | 249 | 100 | 21–100 | 100 | 98–100 |

TP: true positives (number of organisms identified correctly by the BCID2 and which were detected with the reference method); FP: false positives (number of organisms which were detected by the BCID2 but not detected according to the reference method); FN: false negatives (number of organisms which were not detected by BCID2 but were detected by the reference method); TN: true negatives (number of organisms which were not detected by BCID2 and which were not present according to the reference method). TP, FP, FN, TN, sensitivities, and specificities were calculated based on the specific target organism. Samples in which a different species was detected/ isolated were considered negative for the purposes of the calculation.

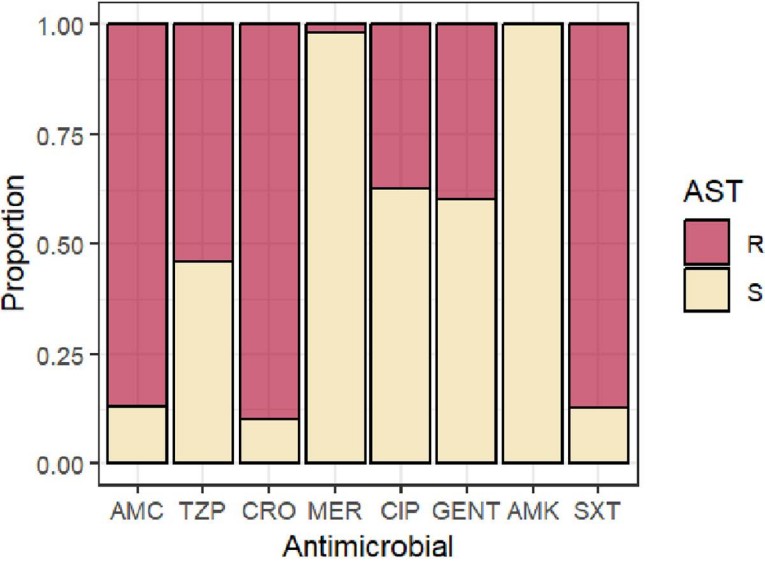

**Fig 3. Antimicrobial susceptibility testing results of *Klebsiella* spp. isolates.** AMC – Amoxicillin-clavulanic acid; TZP – Piperacillin/Tazobactam; CRO – Ceftriaxone; MER – Meropenem; CIP – Ciprofloxacin; GENT – Gentamicin; AMK – Amikacin; SXT – Sulfamethoxazole/Trimethoprim.

## Usability of the BIOFIRE instrument with the BCID2 panel

A total of five laboratory scientists participated in the BCID2 panel system usability assessment. As illustrated in Table 4, the overall SUS score was 79.5 which is higher than the average usability score of 68 which is in use for SUS scoring widely.

## Discussion

Overall BCID2 had a good performance for the identification of most pathogens and for the detection of resistance genes. The test was easy to use and can reduce turnaround times of laboratory testing and improve clinical care.

Rapid and correct identification of pathogens and resistance to commonly used antibiotics is critical given the high burden of third-generation cephalosporin resistant Enterobacterales, predominantly *K. pneumoniae* isolated from neonates with sepsis in this study. Previous studies from this [4] and similar settings [17–19] have shown that time to results for blood cultures often exceeds three days preventing any meaningful impact on clinical decision making. Mortality and morbidity of neonatal sepsis is generally high and effective treatment informed by antimicrobial susceptibly testing is one of the few effective interventions [20].

Overall BCID2 performance was lower than that reported in studies conducted in high-resource settings [11]. This may be due to the pragmatic nature of this study which was implemented in a low-resource laboratory without ready access to accurate identification methods such as MALDI-TOF or sequencing. Hence discrepancies could not be investigated in real-time. Further, blood cultures in this study predominantly grew Enterobacterales other than *E. coli* for which sensitivity may be lower [11]. Of note is that some organisms were only detected in very few samples leading to wide confidence intervals around the sensitivity estimate.

False negative results may be explained by low bacterial load in some samples, and interference by the presence of other organisms within polymicrobial samples. Other authors reported that BCID2 results initially categorised as "false negative" were confirmed as "true negatives" in 38% following confirmatory testing [21]. However, in this study discrepancies were not investigated in real time because of staff shortages and reagent stock outs in the routine laboratory.

**Table 4. BCID2 panel system usability score (N=5).**

| System Usability Scale Statements | Mean score (SD) | Converted* |
|---|---|---|
| 1. I think that I would like to use this system frequently | 3.8 (1.6) | 2.8 |
| 2. I found the system unnecessarily complex | 1.6 (0.5) | 3.4 |
| 3. I thought the system was easy to use | 4.2 (0.4) | 3.2 |
| 4. I think that I would need the support of a technical person to be able to use this system | 1.8 (0.4) | 3.2 |
| 5. I found the various functions in this system were well integrated | 4 (0) | 3 |
| 6. I thought there was too much inconsistency in this system | 1.2 (0.4) | 3.8 |
| 7. I would imagine that most people would learn to use this system very quickly | 3.8 (0.4) | 2.8 |
| 8. I found the system very cumbersome to use | 1.8 (0.4) | 3.2 |
| 9. I felt very confident using the system | 4.6 (0.5) | 3.6 |
| 10. I needed to learn a lot of things before I could get going with this system | 2.2 (0.4) | 2.8 |
| **Calculated SUS score (total converted mean scores X 2.5)** | **79.5** | |

*Converted score out of 4; calculated by the formula x-1 for odd numbered questions and 5-x for even numbered questions where x=mean score.

False positives detected by BCID2 could have resulted from the presence of non-viable bacterial genomic material following antimicrobial treatment, contamination during sampling or the presence of bacterial genomic material in blood culture bottles which has been previously reported [21]. Considering the very high sensitivity of molecular methods, incorporating some quantification measure for pathogen detection by BCID2 could be useful for result interpretation.

One in four blood cultures grew an organism which was classified as a likely contaminant despite a quality improvement programme implemented alongside the study. The quality improvement programme aimed at improving blood culture collection techniques through standard operating procedures shared with junior doctors via electronic messaging groups and as posters throughout the neonatal and paediatric wards and regular training sessions. Factors contributing to high contamination rates included staff shortages, high staff turnover, suboptimal equipment, particularly sterile blood culture collection packs, and lack of consumables. Also sufficient volumes from venepuncture, which are crucial to avoid false negative results, and using aseptic technique is challenging to perform in neonates and young children which made up the majority of the patient population in this study. While the contamination rates were unacceptably high, this reflects the reality in low resource settings where disease burden and bed occupancy are high, patient to staff ratio low and stock outs of consumables a frequent phenomenon [22].

Given the high contamination rates it is not surprising that a considerable proportion of cultures were polymicrobial as per BCID2 results. Not all cultures identified as polymicrobial by BCID2 were confirmed as such by phenotypic methods. This may partly be explained by the lack of experience and training in interpreting mixed cultures in laboratories without ready access to bacterial identification. BCID2 has been reported to perform less well on polymicrobial samples which may explain why the sensitivity estimates in this study were lower compared to studies conducted in high resource settings [23,24].

The most common off-panel organisms detected were *Bacillus* spp. and *Corynebacterium* spp. These organisms, which are almost always contaminants, are difficult to identify in low-resource settings. In the context of expanded use of molecular methods for pathogen identification from positive blood cultures in low resource settings, inclusion of genus-specific probes for common contaminants should be considered. This would reassure laboratory staff and health care workers and aid in result interpretation and treatment decisions.

In low resource settings where skilled laboratory scientists are scarce, staff turnover and attrition are high and regular training difficult to implement and sustain ease of use is an important factor when deciding whether or not a test should be implemented. The BCID2 assay and system was easy to use and laboratory scientists felt confident in performing the test. Overall, the system achieved a SUS score of 79.5 indicating a good usability in this setting [15,16,25]. These data are encouraging specifically because the assay was performed by routine laboratory staff with only two hours of assay specific training. Results are likely to be generalisable to similar settings (i.e., laboratories housed in public hospitals in low resource settings) and to molecular cartridge-based assays of comparable low-technical complexity.

Currently the use of the BCID2 assay is cost prohibitive even in high resource settings. Costs but more importantly cost-effectiveness are important considerations when considering implementation of the BCID2 or similar assays. Given the low quality of microbiology laboratories in low resource settings rapid molecular assays for pathogen identification and resistance detection are likely to have a significant effect on patient-important outcomes, but no efficacy and effectiveness data are yet available.

According to the WHO essential diagnostic list blood culture diagnostics and antimicrobial susceptibility testing should be available in clinical laboratories [26]. Recognising the potential

for simplifying laboratory workflows and reducing turnaround times, the need for evaluating molecular tests for rapid pathogen identification and resistance detection from blood cultures has been highlighted as one of the WHO top priorities for AMR research [10]. Rapid molecular multiplex assays with low level of complexity could potentially be a game-changer for clinical management of sepsis if supplemented by or integrated with a limited battery of other tests (in parallel or sequentially). For example the rapid detection of ESBL-*K. pneumoniae* by BCID2 in our setting is important to inform discontinuation of ceftriaxone. However, the BCID2 results are insufficient to guide appropriate antimicrobial treatment. Hence in the absence of sequencing which allows prediction of resistance and susceptibility of a large number of antimicrobials, molecular tests will need to be done in parallel with a selected number of phenotypic antimicrobial susceptibility tests. The phenotypic testing should be limited to available, affordable and commonly used antimicrobials, e.g., aminoglycosides, fluoroquinolones and chloramphenicol for Gram-negatives.

### Limitations

The study was limited by the lack of access to real-time bacterial identification to investigate discrepancies, and by low numbers of pathogens such as *S. pneumoniae* and *N. meningitidis* and resistance determinants detected. Further, we were not able to investigate the impact of rapid testing on antimicrobial prescriptions and patient important outcomes.

## Conclusion

This is one of the few studies evaluating rapid pathogen identification and resistance detection from positive blood culture in a low-resource setting and shows that rapid molecular tests of low-technical complexity are promising for decreasing turnaround time and improving the quality of diagnostics for patients with sepsis.

### Recommendations

Setting specific workflows will need to be developed to supplement molecular tests with a minimal battery of phenotypic tests to ensure optimal clinical decision making. The effect and cost-effectiveness of improved diagnostic workflows with rapid molecular diagnostics at its core on patient important outcomes will need to be investigated.

### Supporting information

**S1 Checklist. Checklist for Standards for Reporting Diagnostic Accuracy**.
(PDF)

**S1 Data. Study dataset and codebook.**
(XLSX)

**S1 Fig. Overview of collected blood cultures samples**.
(DOCX)

**S2 Fig. Number of organisms identified by Biofire according to department for all 780 samples tested using BCID2.**
(DOCX)

**S1 Table. Distribution of organisms detected by BCID2 according to ward for all isolates tested.**
(DOCX)

**S2 Table. Biofire off-panel organisms identified in blood culture positive, BCID2 negative samples.** *9 cultures also had Staphylococcus sp. (n=3), Klebsiella sp (n=5), Enterobacter sp (n=1) which were missed by BCID2.
(DOCX)

**S3 Table. Status of false positive results: detection by BCID2 without identification of the organism using the reference methods.** *Excludes off-panel organisms; samples may contain none, one, or multiple organisms; ACBC: A. calcoaceticus-baumannii complex.
(DOCX)

**S4 Table. Status of false negative results: identification using the reference methods without detection by BCID2.** *Samples may contain none, one, or multiple organisms.
(DOCX)

## Acknowledgments

We would like to extend our gratitude to the research team for implementing the study protocol. We would like to thank the Sally Mugabe Central Hospital Microbiology Laboratory scientists for conducting blood culture testing. Additionally, we thank the laboratory scientists from the National Reference Laboratory (Zimbabwe), IHMA Europe (Switzerland), Ares Genetics GmbH (Austria) and Microbes Ng (United Kingdom) for conducting further tests on the study isolates. Importantly, we would like to thank the clinicians who supported and participated in the blood culture collection training.

## Author contributions

**Conceptualization:** Tinashe Mwaturura, Birgitta Gleeson, Felicity Fitzgerald, Katharina Kranzer.

**Data curation:** Tinashe Mwaturura, Ioana D. Olaru.

**Formal analysis:** Ioana D. Olaru.

**Funding acquisition:** Birgitta Gleeson, Cecilia Ferreyra.

**Investigation:** Jackie Katunga, Tapfumaneyi Mashe.

**Methodology:** Tinashe Mwaturura, Birgitta Gleeson, Cecilia Ferreyra, Katharina Kranzer.

**Project administration:** Tinashe Mwaturura, Seyi Gansallo.

**Resources:** Felicity Fitzgerald, Cecilia Ferreyra, Katharina Kranzer.

**Supervision:** Gwendoline Chimhini, Mutsa Bwakura-Dangarembizi, Belinda Sado, Christopher Pasi, Felicity Fitzgerald, Cecilia Ferreyra, Katharina Kranzer.

**Validation:** Tinashe Mwaturura, Ioana D. Olaru, Katharina Kranzer.

**Visualization:** Ioana D. Olaru.

**Writing – original draft:** Tinashe Mwaturura, Ioana D. Olaru.

**Writing – review & editing:** Gwendoline Chimhini, Mutsa Bwakura-Dangarembizi, Marcia Mangiza, Simbarashe Chimhuya, Belinda Sado, Jackie Katunga, Andrew Tarupiwa, Agnes Juru, Tapfumaneyi Mashe, Christopher Pasi, Veronicah Chuchu, Seyi Gansallo, Birgitta Gleeson, Felicity Fitzgerald, Cecilia Ferreyra, Katharina Kranzer.

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
