## [Decision Letter · Decision Letter 0]

16 Dec 2024

PGPH-D-24-02035

Rapid bacterial identification and resistance detection in positive blood cultures from patients with sepsis in Harare, Zimbabwe: evaluation of a low complexity molecular diagnostic platform

Dear Dr. Mwaturura,

Thank you for submitting your manuscript to PLOS Global Public Health. After careful consideration, we feel that it has merit but does not fully meet PLOS Global Public Health’s publication criteria as it currently stands. Therefore, we invite you to submit a revised version of the manuscript that addresses the points raised during the review process.

We look forward to receiving your revised manuscript.

Kind regards,

Shivanthi Samarasinghe, PhD

Academic Editor

Journal Requirements:

1. We ask that a manuscript source file is provided at Revision. Please upload your manuscript file as a .doc, .docx, .rtf or .tex.

2. Please provide separate figure files in .tif or .eps format.

3. We do not publish any copyright or trademark symbols that usually accompany proprietary names, eg (R), (C), or TM  (e.g. next to drug or reagent names). Please remove all instances of trademark/copyright symbols throughout the text, including ® on pages 2, 4, 6, 22.

Additional Editor Comments (if provided):

PGPH-D-24-02035

"Rapid bacterial identification and resistance detection in positive blood cultures from patients with sepsis in Harare, Zimbabwe: evaluation of a low complexity molecular diagnostic platform."

The study paves a pathway to participating in solving many technical problems associated with bacterial identification.

Considering the comments review comments, I’m delighted highlighted publication can be accepted with minor revision.

Please address the following comments given by reviewers:

Reviewer 1 comments;

Title:

Suggestion ( Rapid bacterial identification and resistance detection using evaluation of a low complexity molecular diagnostic platform).

Abstract:

1. write in separate subtitle of abstract background, aim or objective of the study, methods, results, conclusions, and results. Write meanfull and comprehensive key words of the research paper. Then author can add abbreviations.

Introduction:

2. The information in introduction can discussed in details that include different detection methods in low and middle income countries compared with the low one. also the stages of development from past to date.

3. If the author use these methods for detection of blood stream infections or sepsis, the author can write these methods in methodology paragraph in details.

4. These information can discussed in details that include different detection methods in low and middle income countries compared with the low one. also the stages of development from past to date (line 67-75).

5. If the author use these methods for detection of blood stream infections or sepsis, the author can write these methods in methodology paragraph in details(line 67-75).

Methodology:

6. Write about type of the study and study participants.

7. Show what are the inclusion and exclusion criteria of the study participants.

8. Enough to point to the specificity of the hospital only

9. off point (line 89-91)

10. write about blood culture diagnostic principals

11. mention the reactant and the product of the tests instead (line 103)

12. Then mention the reactant and the product of the tests instead (line 105-110)

13. Rearrange data (line 125-148)

-These data for statistical analysis.

add the title below:

-preparation date for analysis.

then:

-data analysis.

-repeated data. add these data to study participant (system usability group)(line 150-151)

-better to write one principle for methodology which include reagent and system.(152-163)

14. Those 5 participants who are staff member of the lab; author can add them as one of the study groups.

Discussion:

15. please discus the results of current study in more details.

16. put the results in categorized paragraphs include result of current study, discussion of result then comparison with other globally researches.

17. Avoid repeating data and information.

18. Avoid writing in self expression., and do not confirm the result by your self according to the high quality of techniques of staff, but confirming of results is by the matching and comparing the current results with previous studies.

19. Better to write the active ingredient of all antibiotics(line 270)

20. Delete the repeated data (line 355-358)

21. Put discussion, conclusion, recommendations and limitations all in separated paragraph.

22. Summarize the acknowledgment

Reviewer 2

1. The title is long therefore requires rephrasing

2 Under the subsection study setting and participant inclusion

3 it is not clear if participants were enrolled and blood samples taken from them or if samples were stored. The authors need to clearly write that section.

4 Do the readers understand reference testing/ please define what is referred to as reference testing

5. Under the subsection of data management the statement “Sample size was determined by the available number of BCID2 panel cartridges” seems out of place therefore should be removed .

When submitting, please highlight the changes in the revised manuscript.

Best Wishes

Dr S Samarasinghe

Reviewers' comments:

Reviewer's Responses to Questions

**Comments to the Author**

1. Does this manuscript meet PLOS Global Public Health’s publication criteria ? Is the manuscript technically sound, and do the data support the conclusions? The manuscript must describe methodologically and ethically rigorous research with conclusions that are appropriately drawn based on the data presented.

Reviewer #1: Yes

Reviewer #2: Partly

2. Has the statistical analysis been performed appropriately and rigorously?

Reviewer #1: Yes

Reviewer #2: Yes

3. Have the authors made all data underlying the findings in their manuscript fully available (please refer to the Data Availability Statement at the start of the manuscript PDF file)?

Reviewer #1: Yes

Reviewer #2: Yes

4. Is the manuscript presented in an intelligible fashion and written in standard English?

Reviewer #1: No

Reviewer #2: Yes

5. Review Comments to the Author

Reviewer #1: Overview:

This study is one of the recent study and it give an excellent impression that it may participating in solving many technical problems.

Title:

Suggestion ( Rapid bacterial identification and resistance detection using evaluation of a low complexity molecular diagnostic platform).

Abstract:

1. write in separate subtitle of abstract background, aim or objective of the study, methods, results, conclusions, and results. Write meanfull and comprehensive key words of the research paper. Then author can add abbreviations.

Introduction:

2. The information in introduction can discussed in details that include different detection methods in low and middle income countries compared with the low one. also the stages of development from past to date.

3. If the author use these methods for detection of blood stream infections or sepsis, the author can write these methods in methodology paragraph in details.

4. These information can discussed in details that include different detection methods in low and middle income countries compared with the low one. also the stages of development from past to date (line 67-75).

5. If the author use these methods for detection of blood stream infections or sepsis, the author can write these methods in methodology paragraph in details(line 67-75).

Methodology:

6. Write about type of the study and study participants.

7. Show what are the inclusion and exclusion criteria of the study participants.

8. Enough to point to the specificity of the hospital only

9. off point (line 89-91)

10. write about blood culture diagnostic principals

11. mention the reactant and the product of the tests instead (line 103)

12. Then mention the reactant and the product of the tests instead (line 105-110)

13. Rearrange data (line 125-148)

-These data for statistical analysis.

add the title below:

-preparation date for analysis.

then:

-data analysis.

-repeated data. add these data to study participant (system usability group)(line 150-151)

-better to write one principle for methodology which include reagent and system.(152-163)

14. Those 5 participants who are staff member of the lab; author can add them as one of the study groups.

Discussion:

15. please discus the results of current study in more details.

16. put the results in categorized paragraphs include result of current study, discussion of result then comparison with other globally researches.

17. Avoid repeating data and information.

18. Avoid writing in self expression., and do not confirm the result by your self according to the high quality of techniques of staff, but confirming of results is by the matching and comparing the current results with previous studies.

19. Better to write the active ingredient of all antibiotics(line 270)

20. Delete the repeated data (line 355-358)

21. Put discussion, conclusion, recommendations and limitations all in separated paragraph.

22. Summarize the acknowledgment

Reviewer #2: 1 The title is long therefore requires rephrasing

2 Under the subsection study setting and participant inclusion

3 it is not clear if participants were enrolled and blood samples taken from them or if samples were stored. The authors need to clearly write that section.

4 Do the readers understand reference testing/ please define what is referred to as reference testing

5. Under the subsection of data management the statement “Sample size was determined by the available number of BCID2 panel cartridges” seems out of place therefore should be removed .

6. PLOS authors have the option to publish the peer review history of their article (what does this mean? ). If published, this will include your full peer review and any attached files.

**Do you want your identity to be public for this peer review?** For information about this choice, including consent withdrawal, please see our Privacy Policy .

Reviewer #1: **Yes: ** Nahla Ahmed Mohammed Abderhman

Reviewer #2: **Yes: ** ESTER LILIAN ACEN

---

## [Decision Letter · Decision Letter 1]

7 Feb 2025

Rapid bacterial identification and resistance detection using a low complexity molecular diagnostic platform in Zimbabwe

PGPH-D-24-02035R1

Dear Mwaturura,

We are pleased to inform you that your manuscript 'Rapid bacterial identification and resistance detection using a low complexity molecular diagnostic platform in Zimbabwe' has been provisionally accepted for publication in PLOS Global Public Health.

Best regards,

Shivanthi Samarasinghe, PhD

Academic Editor

Reviewer Comments (if any, and for reference):

Reviewer's Responses to Questions

**Comments to the Author**

1. If the authors have adequately addressed your comments raised in a previous round of review and you feel that this manuscript is now acceptable for publication, you may indicate that here to bypass the “Comments to the Author” section, enter your conflict of interest statement in the “Confidential to Editor” section, and submit your "Accept" recommendation.

Reviewer #3: All comments have been addressed

2. Does this manuscript meet PLOS Global Public Health’s publication criteria ? Is the manuscript technically sound, and do the data support the conclusions? The manuscript must describe methodologically and ethically rigorous research with conclusions that are appropriately drawn based on the data presented.

Reviewer #3: Yes

3. Has the statistical analysis been performed appropriately and rigorously?

Reviewer #3: Yes

4. Have the authors made all data underlying the findings in their manuscript fully available (please refer to the Data Availability Statement at the start of the manuscript PDF file)?

Reviewer #3: Yes

5. Is the manuscript presented in an intelligible fashion and written in standard English?

Reviewer #3: No

6. Review Comments to the Author

Reviewer #3: The authors have addressed all the reviewer comments thoroughly. However, there are several sentences with grammatical errors and incorrect English, which may affect the clarity of the manuscript. Additionally, some sentences are difficult for readers to understand.

I have highlighted these areas in yellow, with suggested corrections provided in blue within the manuscript.

Considering these minor issues, I recommend accepting the manuscript with minor revisions as suggested.

7. PLOS authors have the option to publish the peer review history of their article (what does this mean? ). If published, this will include your full peer review and any attached files.

**Do you want your identity to be public for this peer review?** For information about this choice, including consent withdrawal, please see our Privacy Policy .

Reviewer #3: No
